# Therapeutic Efficacy of ABN401, a Highly Potent and Selective MET Inhibitor, Based on Diagnostic Biomarker Test in *MET*-Addicted Cancer

**DOI:** 10.3390/cancers12061575

**Published:** 2020-06-15

**Authors:** Jooseok Kim, Kyung Eui Park, Yoo-Seong Jeong, YeongMun Kim, Hayeon Park, Ji-Hye Nam, Kyungsoo Jung, Woo Sung Son, Hun Soon Jung, Jong-Hwa Lee, Seong Hoon Jeong, Nam Ah Kim, Jae Du Ha, Sung Yun Cho, Yoon-La Choi, Suk-Jae Chung, Jun Young Choi, Sungyoul Hong, Young Kee Shin

**Affiliations:** 1College of Pharmacy and Research Institute of Pharmaceutical Sciences, Seoul National University, Seoul 08826, Korea; ansrkd5@snu.ac.kr (J.K.); jus2401@snu.ac.kr (Y.-S.J.); sukjae@snu.ac.kr (S.-J.C.); sungyoul@snu.ac.kr (S.H.); 2R&D Center, ABION Inc., Seoul 08394, Korea; pku1218@abionbio.com (K.E.P.); yeongmun@abionbio.com (Y.K.); candybk@abionbio.com (H.P.); hunsoonjung@enhancedbio.com (H.S.J.); namah87@dongguk.edu (N.A.K.); comfly@abionbio.com (J.Y.C.); 3Department of Manufacturing Pharmacy, College of Pharmacy, Seoul National University, Seoul 08826, Korea; 4Department of Molecular Medicine and Biopharmaceutical Sciences, Graduate School of Convergence Science and Technology, Seoul National University, Seoul 08826, Korea; micorazon12@logonebio.org; 5Department of Pathology & Translational Genomics, Samsung Medical Center, Sungkyunkwan University School of Medicine, Seoul 06351, Korea; jks850820@naver.com (K.J.); yunachoi2468@gmail.com (Y.-L.C.); 6Department of Pharmacy, College of Pharmacy and Institute of Pharmaceutical Sciences, CHA University, Gyeonggi-do 11160, Korea; nmrson@gmail.com; 7DMPK Group, Korea Institute of Toxicology, Daejeon 305343, Korea; jhl@kitox.re.kr; 8Lab of Pharmaceutical Engineering, College of Pharmacy, Dongguk University, Gyeonggi 10326, Korea; shjeong@dongguk.edu; 9Bio & Drug Discovery Division, Korea Research Institute of Chemical Technology, Yuseong-gu, Daejeon 34114, Korea; jdha@krict.re.kr (J.D.H.); sycho@krict.re.kr (S.Y.C.); 10Department of Health Sciences and Technology, SAIHST, Sungkyunkwan University, Seoul 06351, Korea

**Keywords:** c-MET, c-MET inhibitor (ABN401), diagnostic biomarker, pharmacokinetics, pharmacodynamics, patient-derived xenograft (PDX) models, gastric cancer, non-small cell lung cancer (NSCLC)

## Abstract

The receptor tyrosine kinase c-MET regulates processes essential for tissue remodeling and mammalian development. The dysregulation of c-MET signaling plays a role in tumorigenesis. The aberrant activation of c-MET, such as that caused by gene amplification or mutations, is associated with many cancers. c-MET is therefore an attractive therapeutic target, and inhibitors are being tested in clinical trials. However, inappropriate patient selection criteria, such as low amplification or expression level cut-off values, have led to the failure of clinical trials. To include patients who respond to MET inhibitors, the selection criteria must include *MET* oncogenic addiction. Here, the efficacy of ABN401, a MET inhibitor, was investigated using histopathologic and genetic analyses in *MET*-addicted cancer cell lines and xenograft models. ABN401 was highly selective for 571 kinases, and it inhibited c-MET activity and its downstream signaling pathway. We performed pharmacokinetic profiling of ABN401 and defined the dose and treatment duration of ABN401 required to inhibit c-MET phosphorylation in xenograft models. The results show that the efficacy of ABN401 is associated with MET status and they highlight the importance of determining the cut-off values. The results suggest that clinical trials need to establish the characteristics of each sample and their correlations with the efficacy of MET inhibitors.

## 1. Introduction

The MET receptor tyrosine kinase consists of an α-chain and a β-chain. MET binds to one ligand, hepatocyte growth factor (HGF), which is under autocrine or paracrine regulation in mesenchymal stromal cells [1,2]. Upon HGF binding, Y1234 and Y1235 within the activation loop of the MET kinase domain are autophosphorylated, leading to the phosphorylation and activation of the multisubstrate docking sites, Y1349 and Y1356. Various adapter proteins, such as GAB1, GRB2, SHC, and SRC, bind to the phosphorylated docking sites of MET, which triggers downstream signaling through MET [3]. MET plays a role in embryonic development, morphogenesis, and tissue repair in normal cells [4,5]. However, *MET* is amplified and/or mutated and/or HGF is overexpressed in cancers [6,7,8,9]. MET promotes tumor invasion, metastasis, and aberrant proliferation, and functions as an activating signal in cancer cells [10,11,12,13]. In gastric cancer, >10% of patients show *MET* amplification and approximately 40% show MET protein overexpression [14,15,16,17]. In non-small cell lung cancer, 2–4% of patients show *MET* amplification and 4% show *MET* mutations including *MET* exon14 deletion [18,19,20,21,22]. *MET* amplification and protein overexpression cause ligand-independent activation, and this is called *MET*-addiction [23,24,25]. MET has therefore emerged as an attractive therapeutic target, and several agents against MET are currently being tested in clinical trials. Although many tyrosine kinase inhibitors against MET have been developed, the first generation MET inhibitors mostly failed because of renal toxicity. This is caused by the metabolism of the 2-quinoline structure of MET inhibitors through aldehyde oxidase, which results in the formation of insoluble metabolites and leads to obstructive toxicity through the formation of crystal structures [26,27,28]. Second generation MET inhibitors were therefore designed by modifying the 2-quinolinone ring structure. However, some clinical trials of second generation MET inhibitors failed because of inappropriate patient selection criteria, such as a low cut-off for *MET* amplification levels. To select patients who respond to MET inhibitors, the selection criteria should include *MET* oncogenic addiction [23,25,29,30]. 

In this study, the efficacy of the highly potent and selective MET tyrosine kinase inhibitor, ABN401, was investigated using histopathologic and genetic analyses (e.g., immunohistochemistry (IHC), fluorescence in situ hybridization (FISH), next-generation sequencing (NGS), and quantitative real-time PCR (q-PCR)) in *MET*-addicted cancer cell lines and patient-derived xenograft (PDX) models. ABN401 was highly selective for MET among 571 kinases (369 wildtype kinases and 202 kinase mutants) and showed >90% cytotoxicity in *MET*-addicted cancer cells. The therapeutic responses were associated with the inhibition of both the autophosphorylation of the MET kinase domain and its downstream signals. ABN401 showed antitumor activity in cancer cell xenografts and PDX models. ABN401 demonstrated favorable pharmacokinetic (PK) properties in Sprague–Dawley (SD) rats, beagle dogs, and cynomolgus monkeys, and the dosage and treatment durations of ABN401 required to inhibit c-MET activity were determined [31]. The present results highlight the importance of selecting a *MET*-driven population in clinical trials. ABN401 is currently being tested in a phase I/II trial in Australia (NCT04052971) and received Investigational New Drug (IND) approval in June 2019 in South Korea.

## 2. Results

### 2.1. Binding Modes of ABN401 Determined by Molecular Docking Simulation

A compound with pyridoxazine as a binder to the hinge region of the ATP binding site of the MET tyrosine kinase was synthesized and designated as ABN401, as shown in Figure 1A. ABN401 was selected as a potent inhibitor of the tyrosine kinase activity of MET. The potential binding mode between ABN401 and the MET kinase domain was investigated using molecular docking methods. ABN401 was docked with the MET kinase domain using PyRx implemented with AutoDock vina. The lowest energy protein–inhibitor complexes were stabilized by hydrogen bonds, hydrophobic interactions, and pi-stacking interactions, as shown in Figure 1B. According to the molecular modeling, the amino acid residues most frequently involved in the binding mode of ABN401 were isoleucine-1084 (1084I), tyrosine-1159 (1159Y), methionine-1160 (1160M), asparate-1164 (1164D), asparagine-1167 (1167N), and tyrosine-1230 (1230Y). The amino acid residues capable of interacting with ABN401 were predicted as follows: valine-1092 (1092V), alanine-1108 (1108A), proline-1158 (1158P), lysine-1161 (1161K), histidine-1162 (1162H), glycine-1163 (1163G), histidine-1174 (1174H), alanine-1180 (1180A), arginine-1208 (1208R), asparagine-1209 (1209N), aspartate-1222 (1222D), alanine-1226 (1226A), aspartate-1231 (1231D), and glutamate-1233 (1233E). Notably, 1230Y in the MET kinase domain was a key residue for inhibitor binding through a pi-stacking interaction in the binding mode of ABN401. Residues 1222D and 1233E were identified as important residues that could form a hydrogen bond and a salt bridge with ABN401 to help stabilize the protein–inhibitor complex, respectively. Comparison of binding modes showed that unique residues, including 1158P, 1161K, 1180A, 1226A, and 1233E, bound to ABN401 through hydrogen bonding and/or hydrophobic interactions, and were rarely involved in MET–ABN401 interactions, as shown in Appendix A. These additional residues appeared to be essential for the binding of ANB401 to a relatively larger area of the active site than that bound by other c-MET inhibitors because of differences in the sizes of the molecules.

### 2.2. ABN401 Is a Highly Selective ATP-Competitive c-MET Inhibitor

The kinase selectivity profile of ABN401 was determined by comparison with 571 kinases, including 369 wildtype kinases and 202 kinase mutants from Reaction Biology Corp, as shown in Figure 1C. A dose of 1 μM ABN401 inhibited only MET kinase (98% inhibition), whereas other kinases were not inhibited by ABN401, except CLK1 and CLK4 (37% and 45% inhibition, respectively). The IC50 value of ABN401 was 10 nM. ABN401 inhibited six of 12 *MET* mutants: P991S, T992I, V1092I, T1173I, Y1235D, and M1250T, as shown in Appendix A. Taken together, these results suggest that ABN401 is a highly potent and selective inhibitor of MET tyrosine kinase.

### 2.3. ABN401 Has Cytotoxic Activity against MET-Addicted Cancer Cells

To assess sensitivity to ABN401 in *MET*-addicted cancer cells, *MET* amplification and/or protein expression was detected in eight cancer cell lines and a normal immortalized cell line using IHC, FISH, and q-PCR. *MET* copy number gains were detected in SNU5, SNU620, Hs746T, MKN45, EBC-1, and H1993 cancer cell lines, but not in the SNU638 cancer cell line, whereas the *MET* gene was not amplified in the normal gastric epithelial cell line, HFE145. All cell lines, except HFE145, showed high-intensity membranous c-MET staining (score 3+). *MET*-high amplified SNU5, SNU620, Hs746T, MKN45, EBC-1, and H1993 cells showed c-MET overexpression; however, c-MET overexpression was not associated with *MET* amplification in SNU638 cells, as shown in Appendix A and Table 1. The cells with MET aberration showed constitutive activation, which is termed oncogenic addiction. The oncogenic-addicted cells showed ligand-independent signaling. The standard WST cell viability assay was used to evaluate the cytotoxic activity of ABN401 in all cells. *MET*-addicted cancer cell lines, but not HFE145 normal cells, showed significantly decreased cell viability in response to <10 nM ABN401, as shown in Figure 2A. The IC50 values of ABN401 were several nM (2~43 nM) in *MET*-addicted cancer cells. No cell killing was observed in c-MET negative cells or in the normal cell line HFE145 in response to 10 μM ABN401, as shown in Table 1. These results indicate that ABN401 efficiently suppresses the growth of *MET*-addicted cancer cells and is a potent MET inhibitor for targeted therapy.

### 2.4. ABN401-Mediated Inhibition of MET Signaling and Downstream Pathway

To investigate the effect of ABN401 on the MET signaling pathway, c-MET phosphorylation and c-MET downstream signaling were examined in SNU5, SNU638, Hs746T (+*MET* exon 14 skipping), EBC-1, and H1993 cells. The cells were treated with ABN401 at several doses, and the lysates were harvested for Western blot analysis. A dose of 100 nM ABN401 completely inhibited auto-phosphorylation at Y1234/Y1235, as well as that of the Y1349/Y1354 docking sites of the MET kinase domain. The phosphorylation of AKT and ERK1/2, which are important for anti-apoptosis, cell survival, and proliferation, was inhibited by ABN401 in *MET*-addicted cancer cells, as shown in Figure 2B–F and Appendix A. In the H1993 cell lines, the phosphorylation of AKT and ERK1/2 was not fully decreased, and it was considered to have a higher IC50 value than other cell lines. This remaining activity of p-AKT and p-ERK can be related to EGFR signaling. Since ABN401 induced the activation of the caspase-associated apoptotic signal pathway, the expression levels of the caspase-3 and PARP-1 were also analyzed. Western blot results showed that caspase-3 and PARP-1 expressions were decreased when treated with over 100 nM of ABN401, while those of the cleavage forms were increased. These results indicate that ABN401 strongly inhibits c-MET activation and downstream signaling, such as apoptosis, survival, and proliferation. 

### 2.5. In Vivo Therapeutic Efficacy of ABN401 in MET-Addicted Cancer Cell Models

To evaluate the in vivo therapeutic efficacy of ABN401, SNU5, EBC-1, and SNU638, tumors were implanted subcutaneously into the flanks of BALB/c-nude mice. The mice were treated orally with 3, 10, or 30 mg/kg of ABN401 when the average tumor volume reached 150–300 mm^3^. ABN401 was administered five consecutive days a week for three weeks. SNU-5, EBC-1 and SNU638 tumor volumes were significantly lower in the mice receiving ABN401 than in those treated with vehicle. ABN401 significantly suppressed SNU-5 tumor growth with a TGI index of 24.47% and 89.49% at doses of 3 and 30 mg/kg, respectively, as shown in Figure 3A and Table 2. ABN401 inhibited EBC-1 tumor growth with a TGI index of 51.26% and 77.85% at doses of 10 and 30 mg/kg, respectively, as shown in Figure 3B and Table 2. Additionally, in the case of SNU638 harboring c-MET-high overexpression, but not *MET* gene amplification, ABN401 suppressed the tumor growth with a TGI index of 65.31% and 78.68% at doses of 10 and 30 mg/kg, respectively, as shown in Figure 3C and Table 2. Taken together, these results indicate that ABN401 significantly suppresses tumor growth in a dose-dependent manner in a mouse xenograft model of *MET*-addicted gastric and lung cancer cells.

### 2.6. Identification of PDX Models with MET Aberration Including MET Exon14 Skipping Mutation

To identify representative PDX models, histopathologic and genomic data from eight PDX models were analyzed using FISH, IHC, and NGS. The *MET*-high amplified PDX models included GA3121, LI0612, and LU2503, which had a *MET* copy number of 14, MET/CEP7 ratio > 5, and IHC intensity 3+. Four gastric PDX models (GA2278, GA0075, GA0152, and GA0046) had a *MET* copy number and MET/CEP7 ratio ≤ 1 and an IHC intensity 0–1+. The lung cancer PDX model LU5381 with *MET* exon14 skipping had a moderate *MET* copy number (~5), MET/CEP7 ratio = 2.05, and IHC intensity 3+. *MET*-high amplified PDX models (copy number >10 and MET/CEP7 ratio >5) showed c-MET overexpression. A moderate *MET* copy number (1–5) was associated with various c-MET expression levels (IHC 1+ or 0), as shown in Appendix A and Table 2.

### 2.7. In Vivo Effect of ABN401 in Patient-Derived Xenograft (PDX) Models with MET Aberration

To examine and compare the efficacy of ABN401 in PDX models harboring different MET aberrations, models with moderate *MET* copy numbers (GA2278, GA0075, GA0152, GA0046, and LU5381) and those with high *MET* copy numbers (GA3121, LI0612, and LU2503) were treated with ABN401. In the *MET*-high copy number and IHC 3+ models, ABN401 significantly suppressed tumor growth, increasing TGI from 99.11% to 109.1% at a dose of 30 mg/kg, as shown in Figure 3D–F and Table 1, whereas in the moderate *MET* copy number models, ABN401 did not inhibit tumor growth, as shown in Figure 3H–K and Table 1. In the LU5381 models with the IHC 3+ and *MET* exon14 skipping mutation, ABN401 suppressed tumor growth with a TGI of 63.09% and 75.47% at doses of 10 and 30 mg/kg, respectively, as shown in Figure 3G and Table 1. Although this model did not have *MET*-high amplification, ABN401 suppressed tumor growth because of *MET* exon14 skipping and IHC 3+. The MET aberrations in the models could have caused *MET*-addiction. These results suggest that ABN401 significantly suppresses tumor growth in PDX models with *MET*-high amplification and IHC 3+ or *MET* exon14 skipping and IHC 3+. *MET* copy number, *MET* exon14 skipping mutation, and protein expression levels affect the efficacy of ABN401.

### 2.8. Pharmacokinetic Studies of ABN401 in SD Rats, Beagle Dogs, and Cynomolgus Monkeys

To assess the PK of ABN401, the plasma concentration–time curve of ABN401 was determined following intravenous and oral administration of the inhibitor in SD rats, as shown in Figure 4A, beagle dogs, as shown in Figure 4B, and cynomolgus monkeys, as shown in Figure 4C. The relevant PK parameters, including bioavailability F%, T_1/2_ (terminal half-life), C_max_ (maximum observed peak concentration), T_max_ (time to reach C_max_), and AUC_inf_ (area under the time–concentration curve from zero to infinity), were calculated by standard noncompartmental analyses, as shown in Table 3. After the administration of the ABN401 capsule form to monkeys, the concentration of the ABN401 capsule form in all control group (Group 5) samples was below the quantification limit (1 ng/mL; data not shown). In cynomolgus monkeys, bioavailability F% was observed to be low (i.e., 0.9–2.1%) at doses ranging from 1–10 mg/kg. In SD rats and beagle dogs, however, bioavailability F% was 42.1–56.2 and 27.4–37.7, respectively. Fold-differences in F%, depending on doses and genders, were up to 1.9 and 1.33 for rats, and 1.26 and 1.38 for dogs, respectively. Presumably on the basis of a lack of dose dependency and/or gender differences in F%, these results suggest a favorable PK profile for ABN401 in SD rats and beagle dogs.

### 2.9. Analysis of the Dose and Treatment Duration of ABN401 Required to Inhibit c-MET Phosphorylation In Vivo (PK/PD Correlation)

ABN401 concentration was detected and quantified by LC/MS/MS in plasma and tumor tissue samples (i.e., for both EBC-1 and SNU5). The limit of quantification ranges were set as low as 3 and 10 ng/mL in plasma and tumor tissue samples, respectively. Temporal profiles of ABN401 in plasma and tumor tissues were obtained from EBC-1 and SNU-5 tumor-bearing mice, as shown in Figure 5A,B. In mice receiving an oral administration of ABN401 at a single dose of 10 mg/kg or 30 mg/kg, the relevant PK parameters were determined using standard moment analyses, as shown in Table 4. The concentration–time relationship of ABN401 in EBC-1 and SNU-5 tumor tissues was determined. Based on the tumor to plasma ratio of the AUC_inf_ for ABN401 (AUC_tumor_/AUC_plasma_), the K_p,ss_ (steady-state tissue-to-plasma partition coefficients) values in EBC-1 tumors at doses of 10 and 30 mg/kg were 3.71 and 4.25, respectively, and in SNU-5 tumors the K_p,ss_ values were 2.76 and 2.66, respectively, suggesting that the drug is readily distributed to tumor tissues. To determine the dose and treatment duration of ABN401 that was required to inhibit c-MET phosphorylation in preclinical models, ABN401 was administered at doses of 10 and 30 mg/kg, and tumor samples were collected at nine time-points, as shown in Appendix A. EBC-1 and SNU5 samples were used to investigate the inhibitory effect of ABN401 on c-MET phosphorylation by IHC and Western blot analysis. For analysis of the concentration–time relationship of PK/PD in EBC-1 and SNU-5, data were integrated, as shown in Figure 5C,D and Appendix A. ABN401 inhibited phosphorylated MET at 0.5–12 h at doses of 10 and 30 mg/kg, whereas MET phosphorylation recovered after 24 h. Based on the free drug hypothesis [32], it was considered that the effective concentration of ABN401 at doses of 10 or 30 mg/kg ranged up to 165 nM (e.g., free fraction in the mouse plasma of 0.07), as shown in Appendix A. Therefore, these results suggest that ABN401 has antitumor activity, primarily achieved by the selective inhibition of c-MET phosphorylation.

## 3. Discussion

*MET* gene amplification, mutation, and overexpression have been reported in various cancers and are strongly implicated in tumor growth and proliferation, tumorigenic transformation, anti-apoptosis effects, resistance mechanisms, angiogenesis, and invasiveness [6,10,11,12,13]. MET aberrant cancers have considerable potential for targeted therapy, and c-MET is a promising clinical target that has been investigated in many clinical trials [7,9]. However, the difficulty of the determination of predictive biomarkers or their cut-off values for appropriate patient selection, have led to the failure of many trials [33]. In this study, the efficacy of ABN401 was investigated in *MET*-addicted cancer cell lines and PDX models of MET-related cancers using histopathologic and genetic analyses (e.g., IHC, FISH, NGS, and q-PCR). We showed that ABN401 is highly potent and selective against c-MET, with > 1000-fold selectivity in a panel of 369 wildtype kinases and 202 kinase mutants. The low off-target activity of ABN401 suggests the safety of the kinase inhibitor. ABN401 was cytotoxic against several *MET*-addicted cancer cell lines and inhibited c-MET phosphorylation, as well as downstream signaling. The effect of ABN401 was examined in cell lines and PDX models with MET aberrations, such as alterations in *MET* copy number, c-MET protein expression, and *MET* mutation by IHC, FISH, and NGS. The in vivo antitumor efficacy of ABN401 was evaluated in several cancer cell xenograft and PDX models. In vitro and in vivo experiments showed that the *MET* copy number, c-MET expression level, and *MET* exon14 skipping mutation affect the efficacy of the c-MET inhibitor ABN401, underscoring the importance of determining a specific cut-off value (*MET*-high copy number and IHC 3+ or IHC 3+ and *MET* exon14 skipping). *MET*-amplified cancers have c-MET overexpression, which is a sensitive indicator of the efficacy of MET inhibitors. However, cancers with no amplification can show MET overexpression and respond to MET inhibitors. In tumors with a cut-off value, ABN401 strongly inhibited c-MET activity and suppressed tumor growth in c-MET-driven models. The preclinical PK studies of ABN401 were performed after IV and oral administration of ABN401 at different doses from 0 to 24 h in rats, dogs, and monkeys with no abnormal observations. To predict the effective dose and time-points of ABN401 required to inhibit c-MET phosphorylation, we analyzed the PK/PD correlation. Based on the tumor to plasma ratio of the AUC_inf_ for ABN401 (AUC_tumor_/AUC_plasma_), the K_p,ss_ values were estimated to be 2.66–4.25, indicating that the inhibitor is readily distributed to tumor tissues. In tumor tissues, ABN401 was retained and inhibited c-MET phosphorylation from 0 to 12 h at doses of 10 and 30 mg/kg.

## 4. Materials and Methods

### 4.1. Compound

ABN401, as shown in Figure 1A, is a novel and selective MET inhibitor, and its molecular structure is CH3-C4H8N2-CH2-C6H4-C4H2N2-C4H7NO-CH2-C4N5H-C3H2N2-CH3 (triazolopyridazine). It is synthesized by the Korea Research Institute of Chemical Technology, Sundia (Shanghai, China) and Olon-ricerca (Concord, OH, USA). ABN401 was dissolved in 100% dimethyl sulfoxide (DMSO; #472301; Sigma-Aldrich, Saint Louis, MO, USA) and diluted in the appropriate medium. In the cell viability test, 0.1% DMSO (10 mM) was used as vehicle or no treatment.

### 4.2. In Vitro Kinase Selective Profiling

ABN401 was tested in a single-dose duplicate mode at a concentration of 1 μM in a panel of 571 human protein kinases, including 368 wildtype and 202 mutant kinases. Staurosporine was used as the control compound at a 10 dose IC50 with 3- or 4-fold serial dilutions starting at 10, 20, or 100 μM using an ATP concentration of 10 μM (Reaction Biology Corp, Malvern, PA, USA).

### 4.3. Molecular Docking Calculation

ABN401 was docked into the active site of the MET kinase domain using PyRx and AutoDock vina [34,35]. Before docking calculation, the crystal structure of the MET kinase domain was obtained from the Protein Data Bank website (PDB ID: 3ZC5) and was modified for docking calculations by adding hydrogens. Energy minimization of the 3ZC5 structure was performed using the AMBER force field with the UCSF Chimera [36,37,38], and the AMBER force field parameters were set to the basic values. In the energy minimization step, the steepest descent (100 steps) and conjugate gradient (100 steps) were used to relieve unfavorable clashes. The structure of the inhibitor was drawn with Marvin Sketch 17.28.00 (2018; ChemAxon) [39], and a 3D structure was generated using the Avogadro software [40], which converts 2D chemical structures into 3D structures and minimizes the energy of structures. PyRx 0.8 with the AutoDock vina package was used to identify the binding modes of ABN401. The input files of the protein and MET inhibitor were generated using PyRx in pdfqf format. The center position of the AutoDock vina search space was 21.1916, 78.9873, and 3.7015 in the x-y-z dimensions, respectively. The sizes of the search dimensions were 24.0089 angstrom (X axis), 24.2608 angstrom (Y axis), and 23.3672 angstrom (Z axis). The exhaustiveness parameter was set to 8 for docking in the grid box, and the positions of the inhibitors bound to the protein were generated along with binding affinities and Root-mean-square deviation (RMSD) scores. Interactions between protein and inhibitors were analyzed using the Protein–Ligand Interaction Profiler (PLIP) [41]. The molecular visualization of structures was achieved using UCSF Chimera and Pymol [42].

### 4.4. Cell Lines

Five gastric cancer cell lines (SNU5, SNU620, SNU638, Hs746T, and MKN45) were obtained from the Korean Cell Line Bank. Normal immortalized cell lines (HFE145) were obtained from Dr. Ashktorab (Howard University, Washington, DC, USA). Two non-small cell lung cancer (NSCLC) cell lines, EBC-1 and H1993, were purchased from the Japanese Research Resources Bank and the American Type Culture Collection, respectively. All cell lines were authenticated by short tandem repeat analyses in Korea Genome Information Institute and were checked for mycoplasma using the e-Myco VALID Mycoplasma PCR Detection Kit (#25245; iNtRon Biotechnology, Inc., Seongnam-si, Korea).

### 4.5. Cell Viability ASSAY

Cells were seeded in 96-well plates at a density of 3000–5000 cells per well in the growth medium, supplemented with 10% fetal bovine serum (#SH30084.03; GE Healthcare Life Sciences, Queensland, AUS). The following day, each well was treated with different concentrations of ABN401 and 10 mM DMSO (control) and was incubated at 37 °C with a 5% CO_2_ incubator for 72 h . For cell viability assays, the growth medium was removed and replaced with 100 µL medium containing 10 µL WST reagent solution (#EZ-3000; DoGen) per well. Cells were incubated for 2 h at 37 °C, and absorbance was measured at 430 nm using a multiplate reader (TECAN; GENios Pro). IC_50_ values were calculated using Softmax Pro 5.2.

### 4.6. Western Blotting

Cells with aberrant c-MET were treated with ABN401 for 72 h and then harvested. The samples were lysed with a cell lysis buffer (#9803; Cell Signaling) containing a protease inhibitor cocktail (#11836153001; Roche, Indianapolis, IN, USA) and PhosSTOP (#04906845011, Roche). The concentration of the lysates was measured using the Pierce BCA Protein Assay kit (#23225; Thermo Fisher, Rockford, IL, USA). The lysates were mixed with equal amounts of 1 × SDS-PAGE loading buffer (#S2002; Biosesang, Seongnam-si, korea.) and resolved in 8% SDS-PAGE gels, followed by a transfer to 0.2 µM immunoblot polyvinylidene difluoride membranes (#1620177; Bio-Rad, Quarry Bay, Hong Kong). The membranes were incubated in a TBS-T solution containing 5% bovine serum albumin (#A9647-100G; Sigma-Aldrich, St. Louis, MO, USA) and skim milk (#232100, BD Biosciences, Sparks, MD, USA) for 1 h followed by sequential incubation in primary and secondary antibodies. The primary antibodies were obtained from Cell Signaling and were as follows: anti-Met (#8198), anti-phospho-Met (Y1234/1235, #3077), anti-phospho-Met (Y1349, #3133), anti-Akt (#4691), anti-phospho-Akt (S473, #4060), anti-Erk1/2 (#4695), anti-phospho-Erk1/2 (T202/Y204, #8544), anti-PARP-1(#9532), and Caspase-3(#9662). Anti-β actin was purchased from Santa Cruz (#47778). The secondary antibodies were obtained from Invitrogen and were goat anti-rabbit (#31460) and goat anti-mouse (#31430). The target proteins were detected with Clarity Western ECL Blotting Substrates (#1705061, Bio-Rad).

### 4.7. MET Copy Number Variation (q-PCR)

To assess the gene expressions in eight cancer cell lines, q-PCR was performed using the Roche LC480 system. The *MET* primer (left: aacaaatgaaaaaggcaattgaa; right: ctcagggtggctattccatc) and universal probe (No. 33, #04687663001) were designed using the Roche Universal Probe Library System Assay Design Center. The reference gene RNase P (#4403326, Thermo Fisher) and LightCycler 480 Probes Master were used by TaqMan Copy number Assay kit. The gDNA isolated from the cells was reactivated by the TaqMan Copy Number Assay (#4400291, Thermo Fisher) [43]. The analysis of the copy number variation (MET copy number/RNase P) was determined by relative quantitation using the comparative Ct method. It was measured by using the Ct difference (ΔCt) between MET and RNas P and was compared to the (ΔCt) values of the samples. The *MET* copy number was calculated to the relative quantity.

### 4.8. Immunohistochemistry (IHC)

To examine the MET expression levels in the five gastric cancer cell lines and five PDX tumors, paraffin blocks were deparaffinized using xylene and an ethanol series (100%, 95%, 80%, and 70%). Next, the sections were washed in distilled water, and endogenous peroxidase blocking was performed with 0.3% hydrogen peroxidase (H_2_O_2_) for 10 min. After washing in distilled water, antigen retrieval was performed in a buffer containing 10 mM Tris, 1 mM EDTA, and 0.03% Tween 20 (pH 9.0) for 30 min in a pressure cooker. A Dako pen was used to draw a border line on the slide, and the sections were blocked with 4% BSA in PBS-T for 30 min to prevent nonspecific binding. After washing, the sections were incubated for 1 h at room temperature (RT) with primary antibody (1:100 dilution) against total MET (CONFIRM anti-MET rabbit monoclonal antibody; SP44 clone, #790-4430; Ventana Medical System, Inc., Tucson, AZ, USA), followed by incubation with HRP-conjugated secondary antibody (1:200) for 30 min at RT. After washing in PBS-T, the sections were developed by incubation with diaminobenzidine (DAB) solution for 7 min and then counterstained with Mayer’s Hematoxylin for 3 min. Finally, the sections were dehydrated and mounted, and then analyzed using a computer image analysis system [9,18].

### 4.9. Fluorescence In Situ Hybridization (FISH)

To examine *MET* copy number variation in gastric cancer cell lines and PDX tumors, paraffin-embedded blocks were deparaffinized using xylene and dehydrated using 100% ethanol. Sections were pretreated using a pretreatment kit at 80 °C for 15 min and then hydrated with water at RT for 3 min. After removing moisture, the sections were incubated in a protease solution at 37 °C for 30 min. Next, the sections were washed in purified water for 3 min at RT and then progressively dehydrated with 70%, 80%, and 100% ethanol for 1 min. Ten microliters of probe (probe: Vysis LSI MET SpectrumRed Probe Kit; reference probe: Vysis CEP7 Spectrum Green Probe; #462359; Abbott) were added to the sections, which were covered with a cover glass and sealed with paper bond. For slide denaturation and hydration, the sections and probes on each slide were incubated with Thermobrite at 73 °C for 5 min, followed by incubation at 37 °C for 18 h. After removing the paper bond, the sections were washed with wash buffer at 74 °C for 2 min and dehydrated at RT for 2 min. Finally, the sections were treated with 10 μL DAPI (4′,6-diamidino-2-phenylindole) and incubated at RT for 10 min before the visualization of MET and CEP7 signals using a microscope [9,44].

### 4.10. In Vivo Efficacy Test in Xenograft Models

Female BALB/c-nude mice (5–6 weeks old) were housed under specific pathogen-free conditions in individually ventilated cages with access to sterilized food and water ad libitum. The animals were maintained on a 12 h light/dark cycle in a temperature- and humidity-controlled animal research facility. All experimental procedures and protocols used for the animal study were approved by the Seoul National University Institutional Animal Care and Use Committee (No. SNU-160307, SNU-161221-1). All efforts were made to minimize the suffering of animals. SNU5, EBC-1, and SNU638 cancer cell lines were harvested and resuspended in 50% Cultrex Basement Membrane Extract. The mice were injected subcutaneously into the flank with viable SNU5, EBC-1 or SNU638 cells at 2 × 10^6^ cells/mouse. When the tumors reached an average volume of 150–300 mm^3^, the mice were randomly divided into vehicle control and treatment groups (six mice per group). ABN401 formulated in 20% PEG400 (#25322-68-3, Sigma-Aldrich) in acetate buffer (pH 4.0) was administered orally every day for 3 weeks. The mice were observed daily and their weights were recorded during the treatment period. The sizes of tumors were measured twice weekly with digital calipers, and the volumes of tumors were calculated with the following formula: volume (mm3) = length (mm) × width (mm)^2^ × 0.5.

### 4.11. In Vivo Efficacy Test in Patient-Derived Xenograft Models

For efficacy studies in PDX mouse models, five primary gastric cancer (GA3121, GA0046, GA0075, GA0152, and GA2278), two NSCLC (LU2503 and LU5381), and one hepatocellular carcinoma (LI0612) xenograft models were established by implanting fresh surgical tumor tissues into immunodeficient mice. The tumors from stock mice bearing human primary tumors were harvested, dissected into fragments, and inoculated into female BALB/c-nude mice. Each mouse was implanted subcutaneously at the right flank with a tumor fragment (P3-5, 2–4 mm^3^ in diameter) to induce tumor development. When the average tumor size reached approximately 150 mm^3^, mice were randomly allocated to three groups, with eight mice per group, and were treated daily with orally administered vehicle or ABN401 at 10 mg/kg or 30 mg/kg. All of the procedures related to animal handling, care, and treatment in this study were performed according to the guidelines approved by the Institutional Animal Care and Use Committee of CrownBio (Nos. E1197-U1601, E1197-U1802, E1197-U1803, and E1197-U1804) following the guidance of the Association for Assessment and Accreditation of Laboratory Animal Care. Tumor measurements determining the T/C ratio (tumor volume in control vs. treated mice) and tumor growth inhibition (TGI; %) were taken as endpoints to determine whether tumor growth was delayed. The tumors were measured twice weekly in two dimensions using a caliper, and the volume was expressed in mm3. Tumor volume [length (mm) × width (mm) 2 × 0.5] was expressed as the median tumor volume ± standard error of the mean (SEM) in the different groups of mice. The T/C value (%) is an indicator of tumor response to treatment and a commonly used antitumor activity endpoint. The TGI (%) values, which were obtained on the last day of treatment and were compared with the vehicle group, were calculated as follows: 1 − (Ti − T_0_)/(Ci − C_0_) × 100%. Ti and Ci are the mean tumor volumes of the treatment and control groups on the measurement day; T_0_ and C_0_ are the mean tumor volumes of the treatment and control groups on day 0.

### 4.12. Pharmacokinetic Studies of ABN401 in Sprague–Dawley Rats, Beagle Dogs, and Cynomolgus Monkeys

Thirty SD rats (24 males and 6 females, approximately 7–9 weeks old, approximately 299–390 g on the dosing day) were purchased from Vital River Laboratory Animal Technology Co., Ltd., and beagle dogs (12 males and 3 females, approximately 12–24 months old, approximately 8–14 kg on the dosing day) were purchased from Beijing Marshall Biotechnology Co., Ltd. The rats and dogs were maintained in air-conditioned animal quarters with alternating 12 h light/dark cycles at a RT of 20–26 °C and a relative humidity of 40–70% before and during the trial. All procedures involving the care and handling of the rats and dogs followed the guidelines of the Pharmaron Institutional Animal Care and Use Committee (Nos. 51604-17-786, 51604-17-785). PK studies were designed to characterize the single-dose intravenous and oral PK profiles of ABN401 in rats and dogs. ABN401 was administered intravenously at 5 mg/kg and orally at 5, 20, and 50 mg/kg in the rats, and 2 mg/kg intravenously and 2, 5, and 10 mg/kg orally in the dogs. The formulation was freshly prepared on the day of dosing. For all intravenous studies, dosing solutions were prepared using the required volume of 5% DMSO/45% PEG400 in 50% distilled water, whereas 20% PEG400 in 0.1 M sodium acetate buffer (pH 4) was used to prepare oral formulations. The formulated samples were stirred at RT for at least 10 min prior to and during dosing. For the assessment of plasma drug concentrations, blood samples were collected at 5, 15, 30, 60, 120, 240, 480, and 1440 min post-dose from each rat and dog. Liquid chromatography, coupled with tandem mass spectrometry (LC-MS/MS), was used to quantify ABN401 in plasma samples from rats and dogs. To assess the PK of ABN401 in the cynomolgus monkey, ABN401 was administered once orally in capsule form at doses of 0, 1, 5, and 10 mg/kg, and 1 mg/kg was administered once intravenously. Blood samples were collected from the cephalic vein or the femoral vein prior to administration and at 0.5, 1, 2, 4, 8, 12, and 24 h after administration in the control group which received vehicle and in the orally treated groups. In the intravenous administration group, blood was collected from the cephalic vein or the femoral vein prior to administration and at 0.083, 0.25, 0.5, 1, 2, 4, 8, and 24 h after administration. The concentrations of ABN401 in the blood were analyzed according to a bioanalytical method [KIT Analytical Procedure: AP_ABN401-NCPP_CBA02 (V1.00)]. All procedures involving the care and handling of the monkeys followed the guidelines of the KIT-Institutional Animal Care and Use Committee (No. MN117002).

### 4.13. Pharmacokinetic Analysis

The plasma concentration–time profiles of ABN401 were analyzed using a non-compartmental method in Phoenix WinNonlin v6.1 (Certara, Princeton, NJ, USA). The linear log trapezoidal algorithm weighting, 1/Y × Y, was used for parameter calculation. The mean PK parameters were calculated in individual animals from each treatment group. Concentrations below the lower limit of quantification (BLLOQ) were excluded from the calculation of PK parameters. The mean values of the PK parameters were calculated using Microsoft Excel.

### 4.14. Pharmacokinetic/Pharmacodynamic Correlation Studies

To analyze the PK/pharmacodynamics (PD) of ABN401 in plasma and tumors, tumor-bearing mice were treated with ABN401 at a dose of 10 or 30 mg/kg. At several time points (0.5, 1, 2, 4, 6, 12, 24, 48, and 72 h), the plasma and tumor samples were collected, and an aliquot (50 μL) of the mouse plasma was mixed with a 4-fold volume (200 μL) of acetonitrile, containing an internal standard (i.e., 200 ng/mL of propranolol), and was vortexed for 5 min for the quantification of ABN401 in the plasma. For the quantification of ABN401 in the tumor tissues, the wet weight of tissues was measured, and the sample was vortexed with a 9-fold volume of acetonitrile containing the internal standard for 5 min. In this dilution, the density of tissues was assumed to be unity. The mixture was then centrifuged at 4 °C for 5 min at 13,200 rpm (5415R; Eppendorf, Hamburg, Germany). The supernatant was collected, and an aliquot was analyzed for ABN401 by LC/MS/MS. To assess the PD of ABN401 in the tumors, the harvested samples were homogenized in liquid nitrogen and then lysed in 1 mL cell lysis buffer (#9806, Cell Signaling) containing a protease inhibitor cocktail (#11836153001, Roche) and PhosSTOP (#04906845011, Roche). The extracted proteins were subjected to Western blotting and IHC. Bands representing total MET and phosho-MET (Y1234/5 and Y1349) were quantified using Amersham Imager 600 software (GE Healthcare Inc., Uppsala, Sweden). The phospho-MET to total MET ratio was calculated at two doses and several time points. The baseline value (100%) was based on the vehicle.

### 4.15. Statistical Analysis

Student’s *t*-test or one-way ANOVA was used for multiple comparisons. * *p* < 0.05, ** *p* < 0.01, and *** *p* < 0.001 were considered statistically significant. Statistical analyses were performed using SSPS 16.0, GraphPad Prism 8.4 software, and Microsoft Excel. In vitro assay and in vivo xenograft data were expressed as the mean ± standard deviation (SD), whereas PDX data were presented as the mean ± SEM.

## 5. Conclusions

ABN401 is a potent and highly selective MET inhibitor in *MET*-addicted cancers and shows a favorable PK profile. The testing of ABN401 in clinical trials may be improved by establishing guidelines for patient selection and determining the effective dosage and treatment durations of ABN401 required to inhibit c-MET activity. ABN401 is currently being tested in a phase I/II study, and its safety, pharmcokinetics, pharmacodynamics, antitumor activity, and exploratory biomarker tests are being examined in advanced solid tumors (NCT04052971). In particular, the clinical trial included exploratory studies that isolate ctDNA (cell-free DNA) from CTCs (Circulating tumor cells) obtained from patient blood using a sorting machine (liquid biopsy), to check *MET* copy number, its expression level, and mutations. Through these studies, we are exploring the best method to select patients who will respond optimally to the MET inhibitor, ABN401.

## Figures and Tables

**Figure 1 cancers-12-01575-f001:**
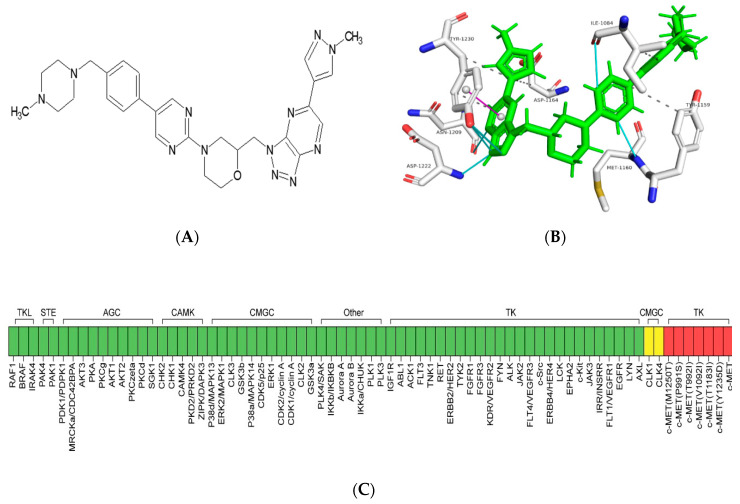
Characteristics of ABN401 as a highly potent and selective c-MET inhibitor. (**A**) Chemical structure of ABN401. (**B**) The binding modes of ABN401 determined using molecular docking simulation. ABN401 in the binding site of the c-MET kinase domain (white colored) is indicated with a line-stick model. The heavy atoms of nitrogen, oxygen, and sulfur are colored in blue, red, and yellow, respectively. The interactions between protein and inhibitors are shown, including hydrophobic interactions (gray colored), hydrogen bonds (cyan colored), and pi-stacking (magenta colored). (**C**) Kinase selectivity profiling of ABN401 in a panel of 571 kinases. TKL: Tyrosine kinase-like, STE: Homologs of yeast Sterile 7, 11, 20 kinases, AGC: Containing PKA, PKG, PKC families, CAMK: Calcium/calmodulin-dependent protein kinase, CMGC: Containing CDK, MAPK, GSK3, CLK families, TK: Tyrosine kinase.

**Figure 2 cancers-12-01575-f002:**
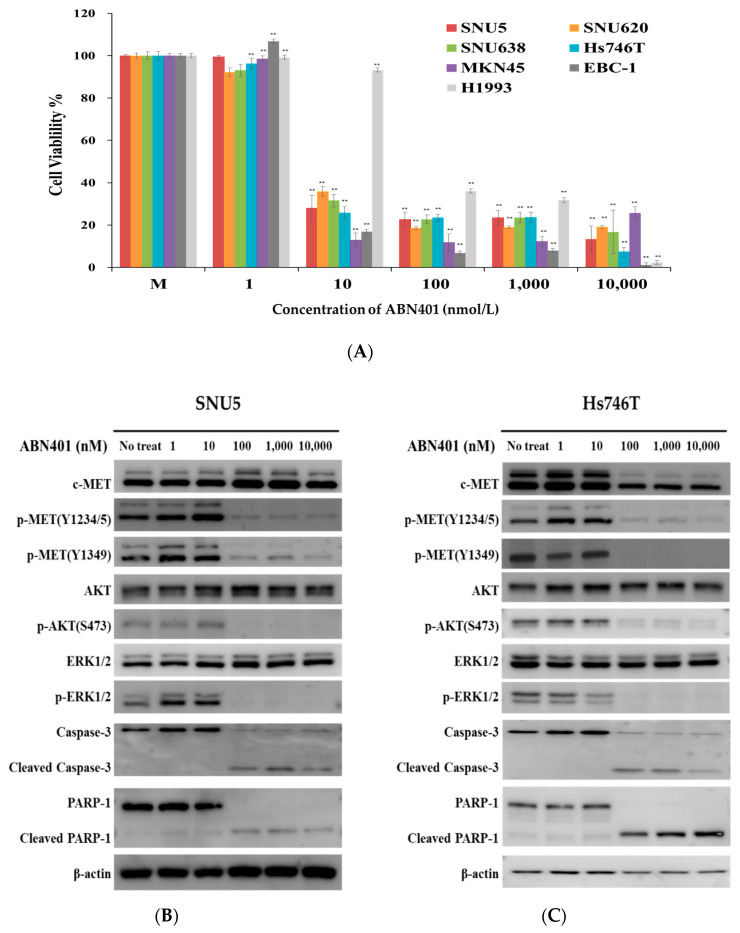
Cytotoxic activity of ABN401 in vitro and effect of ABN401 on MET signaling in *MET*-addicted cancer cells. (**A**) *MET*-addicted cancer cell lines and normal immortalized cells were treated with ABN401 for 72 h, and cell viability was assessed using the WST assay. Data are expressed as the mean ± standard deviation (SD) from three sextuplicate independent experiments. (**B**) SNU5, (**C**) Hs746T, (**D**) EBC-1, (**E**) SNU638, (**F**) H1993 cancer cell lines with MET protein overexpression and/or *MET*-high amplification and/or *MET* exon1 14 skipping treated with different concentrations of ABN401 for 72 h were harvested and examined by Western blotting. Western blotting data are representative of three independent experiments. Comparisons between the control (no treatment group) and treatment groups were performed using the *t*-test. ** *p* < 0.01.

**Figure 3 cancers-12-01575-f003:**
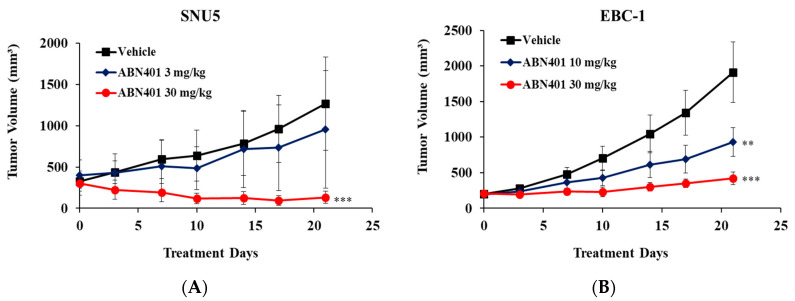
In vivo therapeutic efficacy of ABN401 in *MET*-addicted cancer cell xenograft and c-MET aberrant PDX models. ABN401 suppresses tumor growth in *MET*-addicted (**A**) SNU5, (**B**) EBC-1, and (**C**) SNU638 cell line xenografts, high *MET* copy numbers or c-MET IHC3+ and *MET* exon 14 skipping mutation (**D**) GA3121, (**E**) LI0612, (**F**) LU2503, and (**G**) LU5381 PDX models, but not in moderate *MET* copy number (**H**) GA2278, (**I**) GA0075, (**J**) GA0152, and (**K**) GA0046 PDX models. ABN401 was administered five consecutive days a week for three weeks. For measurement of the tumor growth inhibition index, tumor volumes were measured for three weeks (19 or 22 days) and the results were shown as the mean ± SEM. Comparisons between the vehicle and treatment groups were performed using the *t*-test. * *p* < 0.05, ** *p* < 0.01, *** *p* < 0.001.

**Figure 4 cancers-12-01575-f004:**
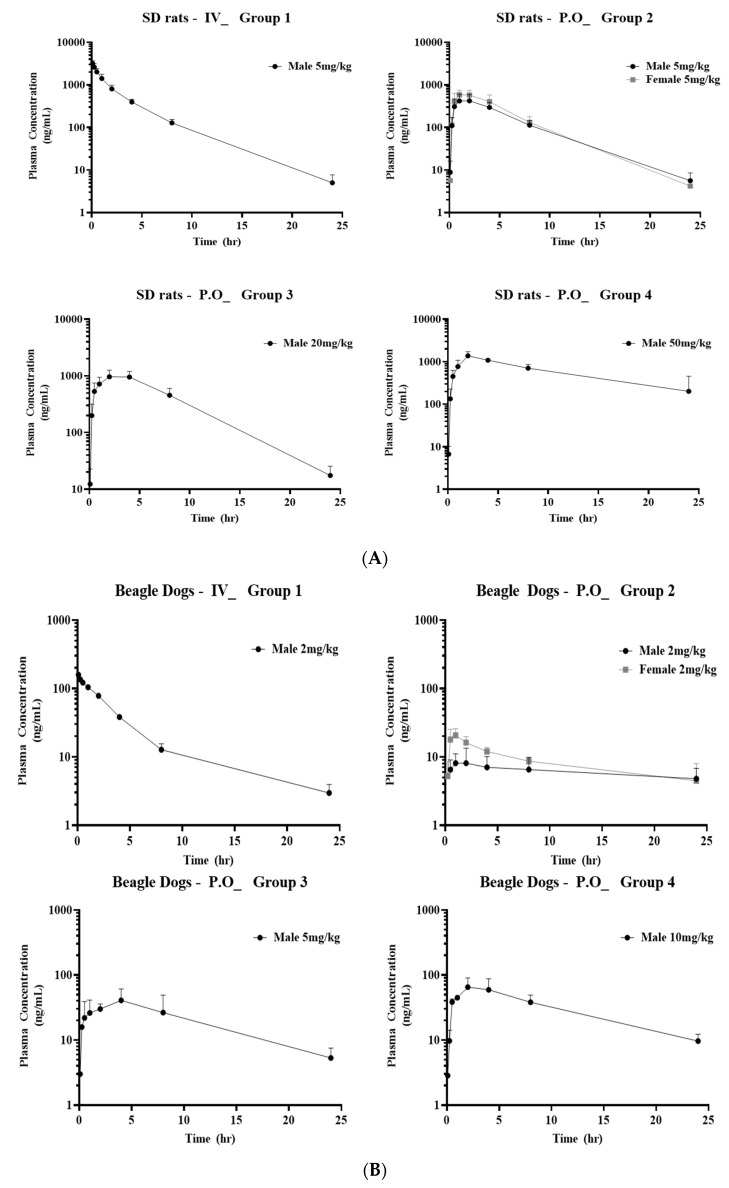
Pharmacokinetics of ABN401. Plasma concentration–time profiles of ABN401 following intravenous and oral administration in (**A**) SD rats, (**B**) beagle dogs, and (**C**) cynomolgus monkeys. The pharmacokinetic profiles including F% (bioavailability), C_0_, C_max_, T_max_, T_1/2_, AUC_0-t_, and AUC_inf_ were measured after intravenous (IV) administration at 5 mg/kg and oral (P.O) administration at 5, 20, or 50 mg/kg in SD rats and IV at 2 mg/kg and oral administration at 2, 5, and 10 mg/kg in beagle dogs. In the cynomolgus monkeys, systemic exposure to ABN401 was detected after a single oral administration at 1, 5, and 10 mg/kg and after a single IV dose of 1 mg/kg.

**Figure 5 cancers-12-01575-f005:**
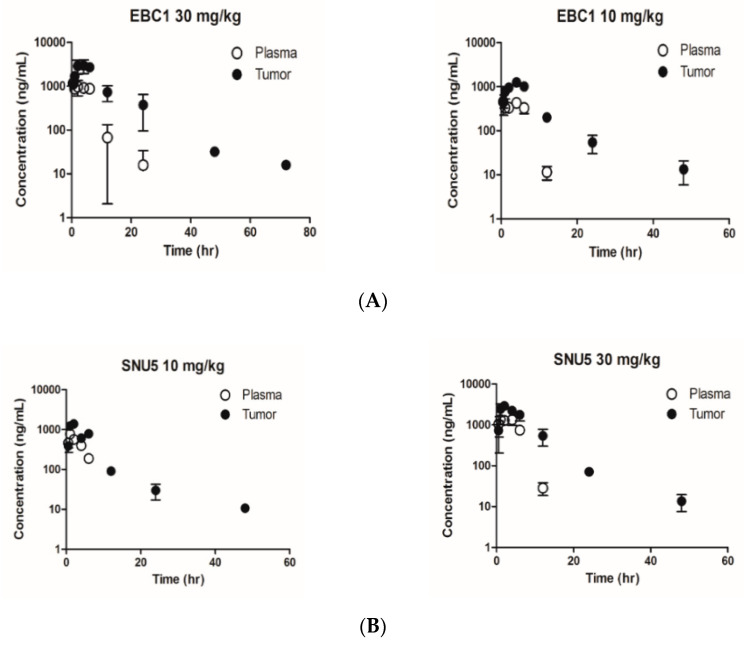
Pharmacokinetic/Pharmacodynamic correlation of ABN401. Quantitative analysis of ABN401 in plasma and tumor tissues in (**A**) EBC-1 and (**B**) SNU-5 models. Based on the tumor to plasma ratio of the AUC_inf_ value for ABN401 (AUC_tumor_/AUC_plasma_), K_p,ss_ values at a dose of 10 and 30 mg/kg of ABN401 were estimated. The dose and duration of ABN401 treatment required to inhibit c-MET phosphorylation were determined by immunohistochemistry (IHC) and Western blotting (WB) in (**C**) EBC-1 and (**D**) SNU-5 models. For analysis of the concentration–time relationship of PK/PD, data were integrated.

**Table 1 cancers-12-01575-t001:** MET status and IC50 value of ABN401 in *MET*-addicted cell lines.

Name	q-PCR	c-MET IHC		*MET* FISH	ABN401 IC_50_ (nM)
	*MET* Copy Number Variation	C	M	MET/CEP7 Ratio	
**SNU5**	18.60	3	3	>5	4.34
**Hs746T**	16.22	3	3	>5	2.00
**MKN45**	17.16	3	3	>5	3.18
**SNU620**	27.1	3	3	>10	8.13
**SNU638**	2.26	3	3	0.8	3.33
**EBC-1**	17.25	3	3	2.31	2.22
**H1993**	22.65	3	3	2.37	43.0
**HFE145** **(Negative cell line)**	1.07	0	0	1	>10,000

C: Cytosol, M: Membrane.

**Table 2 cancers-12-01575-t002:** The MET status and efficacy of ABN401 in cancer cell lines and patient-derived xenograft (PDX) models.

Group	Name	*MET* CNV		c-MET IHC		*MET* FISH	ABN401 Dose (mg/kg)	TGI (%)
		q-PCR	WES *	C	M	MET/CEP7 Ratio		
Xenograft (*MET*-high amplification)	SNU5	18.6		3	3	>5	3	24.47
							30	89.49
	EBC-1	17.25		3	3	>5	10	51.26
							30	77.85
Xenograft (c-MET overexpression)	SNU638	2.26		3	3	0.8	10	65.31
							30	78.68
Patient-derived Xenograft (*MET*-negative)	GA2278		1	0	0	0.8	10	56.0
							30	47.6
Patient-derived Xenograft (*MET*-low amplification)	GA0075		5	0	0	0.6	10	16.2
							30	22.3
	GA0152		5	0	0	0.8	10	7.4
							30	14.2
	GA0046		5	1	1	1	10	17.3
							30	13.4
Patient-derived Xenograft (*MET*-mid amplification)	LU5381 (+*MET* exon14 Skipping)		5	3	3	2.05	10	63.09
							30	75.47
Patient-derived Xenograft (*MET*-high amplification)	LU2503 (+*MET*ex14 skipping)		14	3	3	>5	10	96.75
							30	99.11
	GA3121		14	3	3	>10	10	89.8
							30	102.4
	LI0612		14	3	3	>10	10	86.6
							30	109.1

CNV = copy number variation, q-PCR = quantitative PCR, WES = whole exome sequencing, C = cytosol, M = membrane, TGI = tumor growth inhibition. * *MET* copy number of PDX models was obtained using WES by Crownbio.

**Table 3 cancers-12-01575-t003:** Pharmacokinetics of ABN401 following Intravenous/Oral Administration in (A) SD rats, (B) beagle dogs, and (C) cynomolgus monkey.

Species	Group No.	Dose Level (mg/kg)	Sex	F	C_0_	C_max_	T_1/2_	T_max_	AUC_0-t_	AUC_inf_	MRT_last_
				(%)	(ng/mL *)	(ng/mL)	(hr)	(hr)	(hr X ng/mL)	(hr X ng/mL)	(hr)
Rat	1(IV)	5	M		3520		3.33		5946	5971	3.15
	2(PO)	5	M	42.1		458		2	2501		4.90
			F	56.2		619		1.5	3344		4.55
	3(PO)	20	M	33.6		1019		3	8003		6.14
	4(PO)	50	M	22.2		1428		2	13,219		8.28
Dog	1(IV)	2	M		172		7.37		527	560	4.79
	2(PO)	2	M	27.4		9.20		1	145		11.2
			F	37.7		21.0		1	199		9.15
	3(PO)	5	M	34.5		44.6		4	455		8.37
	4(PO)	10	M	27.5		65.3		2	725		8.68
Monkey	1(IV)	1	M			323.8	0.88	0.083	322.6	327.1	
	2(PO)	1	M			1.9	N/A	1.7	2.9	N/A	
	3(PO)	5	M			7.4	N/A	4.0	33.6	N/A	
	4(PO)	10	M			10.8	6.0	3.3	67.0	78.2	

IV: administration route intravenous, PO: administration route per oral, N/A: not applicable, M: male, F: female, F(%) = bioavailability index, C_0_: after intravenous injection of instantaneous concentration, AUC_last_ = area under the time-concentration curve from zero to the last quantifiable time-point, AUC_inf_ = area under the time-concentration curve from zero to infinity, T_1/2_ = terminal half-life, C_max_ = maximum observed peak concentration, T_max_ = time to reach C_max_, MRT_last_ = mean residence time from zero to the last quantifiable time-point. * 1000 ng/mL = 1765 nM (ABN401 molecular weight = 566.6).

**Table 4 cancers-12-01575-t004:** Quantitative Analysis of ABN401 in the plasma and tumor tissue in mice.

**Plasma**	**10 mg/kg**	**10 mg/kg**	**30 mg/kg**	**30 mg/kg**
	**EBC-1**	**SNU5**	**EBC-1**	**SNU-5**
T_1/2_ (min)	141	153	214	83.8
C_max_ (ng/mL)	456	774	1170	1340
AUC_inf_ (ng·min/mL)	193,000	200,000	527,000	554,000
CL/F (mL/min/kg)	51.8	50.0	56.9	54.1
MRT (min)	255	241	300	230
**Tumor**	**10 mg/kg**	**10 mg/kg**	**30 mg/kg**	**30 mg/kg**
	**EBC-1**	**SNU5**	**EBC-1**	**SNU-5**
T_1/2_ (min)	571	726	522	344
C_max_ (ng/mL)	1260	1390	2950	2920
AUC_inf_ (ng·min/mL)	715,000	553,000	2,240,000	1,470,000

T_1/2_ = terminal half-life, C_max_ = maximum observed peak concentration, T_max_ = time to reach C_max_, AUC_inf_ = area under the time–concentration curve from zero to infinity, MRT = mean residence time, CL/F = oral clearance.

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
