# Peer review of "Therapeutic Efficacy of ABN401, a Highly Potent and Selective MET Inhibitor, Based on Diagnostic Biomarker Test in MET-Addicted Cancer"

_cancers, 2020, doi:10.3390/cancers12061575_

Round 1

Reviewer 1 Report

The manuscript presents interesting data regarding the potential antitumor activities of novel selective c-MET-inhibitor ABN401 in vitro and in vivo. In general, the manuscript is very well-written and the experimental design is well-done too. The data presented in the manuscript is clear and convincing. 

I have the following questions/suggestions regarding the manuscript: 

1) What is the molecular mechanism of cytotoxicity of ABN401 used in this study? And what is an outcome of MET inhibition? Despite very clear data shown on Fig 2A illustrating a dose-dependent cytotoxic activity of MET-inhibitor against the panel of cancer cells lines overexpressing c-MET, data illustrating the expression of apoptotic cell markers on WB on Fig 2B is missing. I would suggest to include the expression of some of apoptotic markers (cleaved caspase-3/PARP, etc) and cell-cycle-related markers (cyclin A2, E2F4, etc.) into the Fig 2B. 

2) The names of the cell lines utilized for in vitro experiments (Fig 2) and xenograft models (Fig 3) do not match well (only 2 lines of 5 shown in Fig.2 were shown for xenograft models). Are the other cell lines (H1933, SNU638) not tumorigenic? Is so, the authors should include in Fig 2A the data obtained on the cell lines and the primary tumors utilized for the xenograft experiments.

3) I suggest to include into the manuscript the IHC-data (if it is currently available) to illustrate whether ABN401 has in an inhibitory impact of Met overactivation in vivo.  Again, IHC-staining for apoptotic markers will make the data more strong and convincing.

Minor: 

4) the data about p<0.05 * shown on the legend (line 186) is missing in figure 3.

5) fig 5 has some technical problems - A, B and C are overlapping in pdf version and unreadable.  

Reviewer 2 Report

Very interesting article, indicating what we already know – the necessity for personalized medicine and molecular testing in cancer patients. However, I have some minor issues: 1. In the article concentration of ABN should be provided either in nM/L or ng/mL to facilitate comparison of provided data. 2. Data provided in 2.2 [113-118] reflects doses of ABN up to 1000 nM, whereas in table 4 the Cmax is up to 2950 ng/mL (up to approx. 6500 nM/L). In my opinion selectivity analyses should be performed with higher concentrations of ABN, which reflects Cmax in tissues. 3. IC50 of ABN is in range 2-4 nM/L (except H1993 in which case is 43 nM/L). Authors should provide some discussion about this fact, especially that H1933 line has the highest cMYC gene copy number and highly positive cMYC in IHC. 4. [170-172] – ABN was administered when tumor reached 150-300 mm3, once daily for 3 weeks, but in [182-184] – ABN was administered 5 days a week until tumors reached a volume of 150–300 mm3. Can you please explain the discrepancy between both statements, if ABN was really administered for 5 days a week, what was the rationale for that? 5. [416] When 415 the size of tumors reached a volume of 150–500 mm3, the mice… previously authors stated that tumor volume before the experiment was 150-300 mm3.

Reviewer 3 Report

Although generally sound, the paper by Kim et al. has some shortcomings; detailed below. 

Some sentences in the text do not make sense, as if the Authors forgot to insert all words. Figure 3 is disrupted and Figure 5 is partly unreadable. The Authors claim that ABN401 caused a cytotoxic effect in c-MET-amplified cell lines. However, they have drawn this conclusion form WST  test, which measures mitochondrial activity. So, the changes in mitochondrial activity may simply reflect changes in proliferation. In order to substantiate the claims of toxicity, other tests measuring toxicity more directly (e.g. cleaved caspase 3) should be used. Initially, seven different cell lines have been tested in vitro. For in vivo studies, two cell lines were chosen arbitrarily without giving any rationale for doing so.   Why are Conclusions placed after Materials and Methods section? Is Appendix really necessary? Maybe those figures can be incorporated as main, or suplementaty?

Round 2

Reviewer 3 Report

I do not have any further questions.

Author Response

The authors are grateful to the reviewer for the helpful notes that we feel have improved the quality of the manuscript.